behaviour

bats, dispersal, *Trachops cirrhosus*, social structure

**Authors for correspondence:**
Victoria Flores
e-mail: vflores@uchicago.edu
Gerald G. Carter
e-mail: carter.1640@osu.edu

†Equal contributions.

# Social structure and relatedness in the fringe-lipped bat (*Trachops cirrhosus*)

Victoria Flores[1,2,†], Gerald G. Carter[2,3,†],

Tanja K. Halczok[4], Gerald Kerth[4] and Rachel A. Page[2]

[1]Committee on Evolutionary Biology, University of Chicago, 1025 E. 57th Street, Chicago, IL 60637, USA
[2]Smithsonian Tropical Research Institute, Apartado 0843-03092, Balboa, Ancón, Republic of Panamá
[3]Department of Evolution, Ecology, and Organismal Biology, The Ohio State University, 318 W. 12th Ave, Columbus, OH 43210, USA
[4]Greifswald University, Zoological Institute and Museum, Soldmannstr. 14, 17489 Greifswald, Germany

 VF, 0000-0002-8021-9787; GGC, 0000-0001-6933-5501; RAP, 0000-0001-7072-0669

General insights into the causes and effects of social structure can be gained from comparative analyses across socially and ecologically diverse taxa, such as bats, but long-term data are lacking for most species. In the neotropical fringe-lipped bat, *Trachops cirrhosus*, social transmission of foraging behaviour is clearly demonstrated in captivity, yet its social structure in the wild remains unclear. Here, we used microsatellite-based estimates of relatedness and records of 157 individually marked adults from 106 roost captures over 6 years, to infer whether male and female *T. cirrhosus* have preferred co-roosting associations and whether such associations were influenced by relatedness. Using a null model that controlled for year and roosting location, we found that both male and female *T. cirrhosus* have preferred roosting partners, but that only females demonstrate kin-biased association. Most roosting groups (67%) contained multiple females with one or two reproductive males. Relatedness patterns and recapture records corroborate genetic evidence for female philopatry and male dispersal. Our study adds to growing evidence that many bats demonstrate preferred roosting associations, which has the potential to influence social information transfer.

## 1. Introduction

Social structure can have profound behavioural and evolutionary consequences, but correctly interpreting social structure requires

understanding the behaviours that drive it. Aggregations can occur passively when individuals are attracted to a common resource, such as a roosting site, or when specific individuals preferentially associate more than expected from mere co-attraction to resources [1,2]. These preferred associations can often occur even in animals demonstrating fission–fusion social dynamics, in which temporary groups of variable size and composition frequently break up and reform (e.g. [3,4]). In addition, dispersal by one sex can lead to kin-biased associations in the philopatric sex [5], and these kin-biased associations can be adaptive when the indirect fitness benefits of kin cooperation are not outweighed by increased kin competition [6].

Many bat species demonstrate non-random social structure including stable social relationships persisting for years, despite frequent roost switching and fission–fusion dynamics (reviewed by [4,7,8]). For example, preferred associations are evident in philopatric female vampire bats, *Desmodus rotundus* [9–11], female Bechstein's bats, *Myotis bechsteinii* [3,12,13], male Jamaican fruit-eating bats, *Artibeus jamaicensis* [14,15], and both male and female Spix's disk-winged bats, *Thyroptera tricolor* [16,17]. In these species, individuals switch between a number of roosts, but emergent social structures can be detected by observing marked individuals over time [4]. In bats, reproduction is slow (usually one or two pups a year) but lifespans can be quite long (up to 40 years), one or both of the sexes disperse at sexual maturity, and kinship within groups typically has a low mean and high variance [4,8,12,18–23]. Taken together, these factors create opportunities for both long-term and kin-biased social relationships, but the degree of kinship bias in association rates varies greatly between species [4]. In many bats, kinship does not seem to be a key driver of co-roosting associations or even cooperative interactions [4,8,12].

Variation in how kinship affects preferred associations in bats provides an opportunity for comparative analyses on the social and ecological causes and consequences of social structure [4], but data from more species are needed for rigorous comparative studies. The fringe-lipped bat, *Trachops cirrhosus*, a member of the ecologically diverse family of leaf-nosed bats, Phyllostomidae, has been studied intensively to understand its foraging behaviour, sensory ecology and social learning [24–28], but the social structure of *T. cirrhosus* is largely unknown.

Fringe-lipped bats are known to day roost in mixed-sex groups of up to 50 individuals in hollow trees, culverts, buildings and caves [29–31]. The mating system of *T. cirrhosus* remains unclear [32]. Females give birth to one offspring at a time coinciding with the start of the rainy season [32], but the gestation length is unknown. During the putative mating season, reproductive males have enlarged testes and create an odorous substance that is smeared on their forearm, called forearm crust [32,33]. Fringe-lipped bats conduct short flights hunting for insects, lizards and frogs, and individuals have home ranges estimated at 60 ha, flying an average of 218 m from their day roost to foraging areas [29]. Captive studies show that individuals learn different prey calls and will learn to associate novel acoustic cues with prey [24]. Novel acoustic cues can also be learned socially and transmitted across individuals [25,27]. Given that social structure can influence both foraging [34] and social information transfer [35,36], the social structure of the fringe-lipped bat is of special interest.

The aim of our study was to infer the social structure of *T. cirrhosus* from roost capture records taken opportunistically over a 6-year period. First, we described the size and sex composition of groups captured at roosts. Second, we used ecologically relevant null models to test whether *T. cirrhosus* have preferred roosting partners, and if so, whether these co-roosting associations were influenced by sex or relatedness. Genetic analyses suggest that male *T. cirrhosus* are the dispersing sex [37], so we predicted that capture data would corroborate genetic evidence for male-biased dispersal. We also predicted that social structure would be driven by kin-biased associations in females that would be evident even after controlling for overlapping individual roost preferences.

# 2. Material and methods

## 2.1. Group captures

Fieldwork was conducted from July 2012 to November 2018. We captured bats using mist nets (Avinet, Dryden, NY) set at the exits of roosts along Pipeline Road and Gamboa Road (figure 1), in Soberanía National Park, Panamá (9.0743° N, 79.6598° W). Roosts were in trees, culverts beneath roads, or in abandoned human-constructed structures in the forest. The Chagres River divides Gamboa Road from Pipeline Road. The Gamboa Road population is genetically differentiated from the Pipeline Road population [37]. When testing the links between age, association and relatedness, we accounted for these population differences in our null models such that effects were tested within each population.

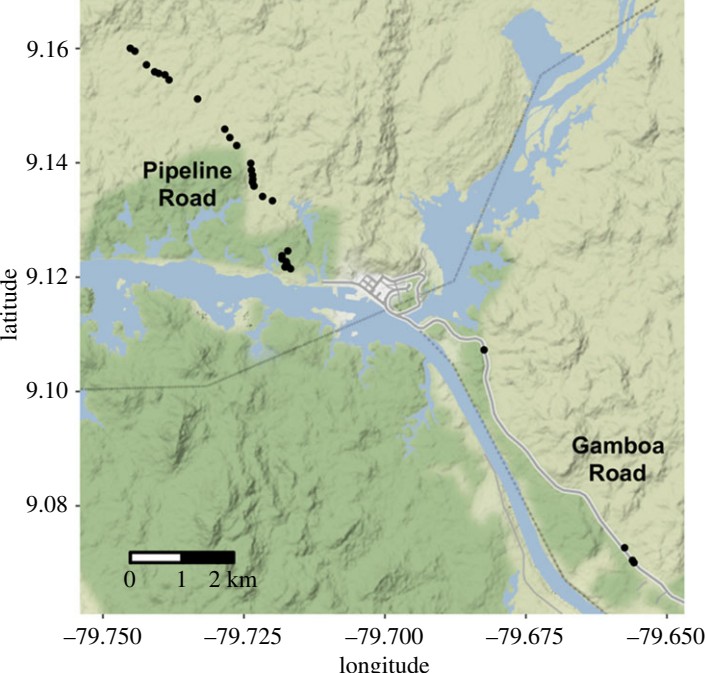

**Figure 1.** Map of study area. Black circles denote roost locations in two populations: along Pipeline Road and along Gamboa Road. Bats never switched roosts between populations.

To remove the effects of parental care on association and relatedness, we focused on adults in our social structure analyses. We identified juveniles by the presence of epiphyseal gaps in the phalanges [38]. For some captures, we also determined reproductive status. We classified females as pregnant by gentle palpation of the abdomen or as lactating by the enlarged size of nipples and the presence of milk, and classified males as reproductive by the scrotal position and presence of enlarged testes [38]. To identify recaptured bats, we marked each bat with a subcutaneous passive integrated transponder tag (Biomark, Boise, ID). All bats were released at their site of capture. Although capture and marking may impact animal movement, *T. cirrhosus* in some populations frequently switch roosts regardless of being captured [29].

## 2.2. Genetic relatedness

We obtained wing tissue samples using a sterilized 4 mm biopsy punch. Tissue samples were stored in 80% ethanol until DNA extraction using an ammonium acetate precipitation method, and bats were genotyped at 16 microsatellite markers [37]. Dyadic relatedness was determined using triadic maximum likelihood in COANCESTRY 1.0.1.5 [39]. See [37] for details.

## 2.3. Group composition

We assessed group composition based on sex and reproductive status. Roosts varied in distance to each other (0.084–0.44 km) and individuals used multiple roosts, so we used a permuted *t*-test (10 000 permutations) to assess whether the mean distances between the roosts used by males and females differed by sex. To estimate a rate of roost switching between known roosts for bats captured four or more times, we calculated the number of individual roosts used by each bat divided by the total number of roost visits by that bat, and then used a permuted *t*-test (10 000 permutations) to assess differences between males and females.

## 2.4. Testing for sex-biased philopatry

To assess evidence for sex-biased philopatry, we used a permuted *F*-test (5000 permutations) to compare the variance in time between the first juvenile capture date and the last capture date as an adult, for males

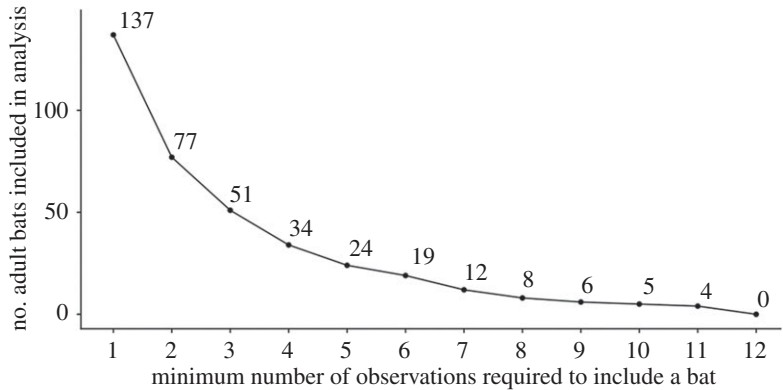

**Figure 2.** Sample sizes of bats observed at least N times. More dyads with bats observed only a few times decreases the precision of dyadic associations, but restricting under-sampled bats reduces the number of individuals.

and females. Male-biased dispersal predicts that young-of-the-year juvenile females, but not males, would be recaptured in later years as adults.

## 2.5. Testing for preferred associations

Similar to previous social network analyses in bats [4], we considered individuals to be associated if they occupied the same day roost at the same time. Although presence in the same space does not always mean individuals are interacting, individuals were frequently clustered in close proximity, often in small culverts less than a metre in diameter. As suggested by Hoppitt & Farine [40], we assessed dyadic association using the simple ratio index, the probability of observing both individuals together given that one has been seen. To assess evidence for preferred associations, we calculated observed and expected social differentiation, the variation in the probability of a dyad being associated, as the coefficient of variation of the simple ratio index [1].

## 2.6. Testing effects of relatedness on association in male dyads, female dyads and female–male dyads

For assessing dyadic relatedness by sex, we used all adult bats. However, we tested our hypotheses about predictors of dyadic association rates using the subset of bats that were captured at least four times. This analysis is preferable because dyadic association rates cannot be precise for any individual seen only a few times [1,2]. For example, association rates are not accurate for bats only seen once or twice. Unlike the estimates of dyadic relatedness, the precision of dyadic association rates is based on small and unequal samples of observations: only 77 of the 137 adult bats were observed more than once, and only five bats were observed at least 10 times (figure 2). We only include bats observed at least four times in our final analyses ($n = 34$), but we also report the results when including all bats.

For comparison with our observed association network, we generated 5000 expected association networks using pre-network permutations [41]. Individuals can be highly associated due to preferring the same roost or being present in the same year. To control for these effects, we constructed our expected (null) networks by repeatedly swapping pairs of individual bat observations within the same population, roost and year. For example, in a given swap, the identity of bat 1 in a particular roost on day 1 in the year 2015 might be swapped with the identity of bat 3 in the same roost on day 4 in that same year. If an observed effect is real, then it should be greater than the null effects from the permuted networks. We calculated one-sided $p$-values as the proportion of null effects more extreme than the observed effect, and then doubled their value to estimate two-sided $p$-values.

To assess whether relatedness differs for same-sex dyads, we constructed three binary matrices (0,1), each encoding the presence of a certain dyad type (female–female, male–male or female–male). We then tested for a correlation with the relatedness matrix using three Mantel tests (5000 permutations), via the vegan R package [42]. We similarly tested for an effect of each dyad type on the association matrix using the quadratic assignment procedure (QAP) in the asnipe R package [43] to compare the three observed beta coefficients with those expected from the random network data (custom null values). Being a female–female dyad or a male–male dyad had opposite effects on association rates (i.e. positive versus

negative coefficients), so we conducted tests of non-random association and kin-biased association separately for adult female–female dyads and male–male dyads. To test for an effect of relatedness on association within each dyad type (female–female, male–male and female–male), we again used QAP in the asnipe R package to compare observed and expected null beta coefficients.

# 3. Results

## 3.1. Size and composition of captured groups

We captured 157 individual bats (69 females, 88 males). Of these, 61 females and 76 males were captured at least once as adults, and 16 females and 18 males were captured at least four times as adults. We did not detect a difference in recapture rates between females ($n = 39$, median = 3, IQR = 3.5, range = 2–12) and males ($n = 46$, median = 3, IQR = 2; range = 2–11, mean difference = −0.88, df = 1, $p = 0.10$). We recorded 106 group captures across 32 different roost sites (415 observations of individual bats). Sizes of captured groups ranged from 1 to 13 individuals (median = 4, IQR = 3). However, these capture numbers were sometimes a subset of actual roosting groups, because individuals sometimes escaped during roost captures. An average of four bats per roosting group is, therefore, an underestimate of true expected group size in roosts. Note that our use of constrained pre-network permutations also control for possible sampling biases, for example, if males are more likely to escape than females [41].

We assessed individuals' reproductive status in the captured groups where reproductive status was recorded, which included 42 mixed-sex groups, 18 all-male groups and three all-female groups. Among the mixed-sex groups, 33 of 42 groups had one reproductive male with forearm crust [32,33], and 9 groups had two reproductive males with forearm crusts. The average relatedness of the two males in these groups (0.08) was not higher than the mean relatedness expected from two random adult males captured at those same times and locations (95% CI: 0.01–0.14). All-male groups always included at least one reproductive male with a forearm crust (median = 2, IQR = 1, range of males in a group = 1–6). We did not detect a difference in the number of males in all-male groups by female reproductive season (permutation test: mean difference = 1.05, $n = 18$, $p = 0.16$). In all eight cases when only one individual was captured in the roost, it was a reproductive male with forearm crust; females were never observed roosting alone.

Bats that were captured at least four times were seen in 3.7 different roosts on average (range = 2–8, median = 3, IQR = 3). We did not detect that adult males used more roosts than adult females (18 males: range = 2–8, median = 4, IQR = 2.25; 16 females: range = 2–7, median = 3, IQR = 3; permutation test: mean difference = 0.22, $n = 34$, $p = 0.67$), or that they disproportionately used roosts that were further apart (permutation test: mean difference = 252 m, $n = 34$, $p = 0.28$).

## 3.2. Sex-biased philopatry

Fifty-two bats were captured as juveniles, and 25 of those were recaptured as adults (males = 13, females = 12). Consistent with male-biased mortality or dispersal, the variation in time between first juvenile capture and last adult capture was greater for females than males (figure 3). Juvenile males were recaptured up to 1.9 years later (range = 195–691 days) while juvenile females were recaptured up to 4.2 years later (range = 165–1514 days, figure 3).

## 3.3. Preferred associations

When controlling for roost and year, preferred roosting partners were detected in adult dyads (CV = 2.54, $n = 34$ bats, $p = 0.0002$), including female–female dyads (CV = 2.27, $n = 16$ bats, $p < 0.0002$) and male–male dyads (CV = 3.10, $n = 18$ bats, $p < 0.0002$). We confirmed that social differentiation was more difficult to detect when including additional bats only observed a few times. When not excluding bats seen less than four times, we could only detect social differentiation in females (137 adults: CV = 5.45, $p = 0.2$; 61 adult females: CV = 4.24, $p = 0.004$, 76 adult males: 6.13, $p = 0.6$).

## 3.4. Effects of relatedness on association

When testing for kin-biased association, we did separate analyses for female–female and male–male dyads because adult female–female dyads were more related (Mantel: $r = 0.04$, 56 bats, $p = 0.009$) and

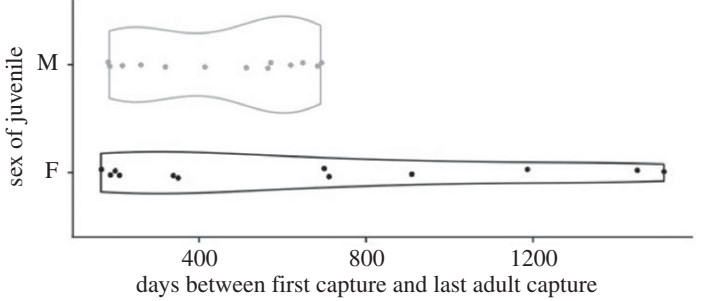

**Figure 3.** Juvenile males were never recaptured as older adults. Females show greater variation in time between the first capture as a juvenile and last capture as an adult ($F = 0.15$, $n = 25$, $p < 0.0001$).

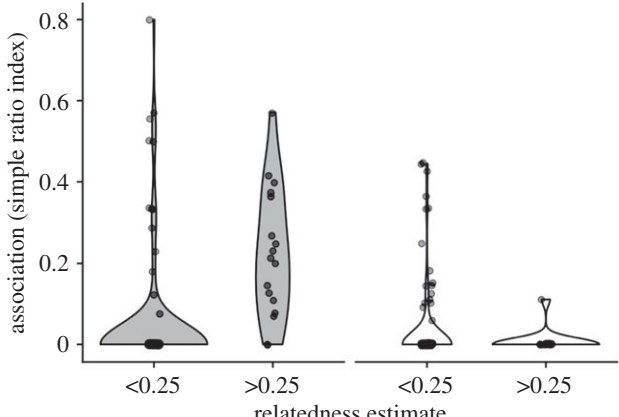

**Figure 4.** Kin-biased association in females but not males. Violin plots show the distributions (probability density function) of association (simple ratio index) among same-sex dyads of females (dark) or males (light) by relatedness. Shaded regions show estimated probability density of the data.

also had higher association rates (QAP beta = 0.03, $p = 0.004$), compared with other dyads. Male–male dyads did not have clearly higher or lower relatedness (Mantel: $r = -0.009$, $n = 67$, $p = 0.7$), but they had lower rates of association (QAP beta = $-0.02$, $p = 0.012$) compared with other dyads.

Relatedness predicted higher association between adults (QAP beta = 0.26, $n = 34$ bats, $p < 0.0001$) and between adult females (QAP beta = 0.43, $n = 16$ bats, $p < 0.0001$), but not between males (QAP beta = $-0.05$, $n = 18$ bats, $p > 0.5$, figure 4). These effects were still evident, but weaker, when we included adults captured fewer than four times.

## 4. Discussion

In this study, we used data from bats captured in their roosts over a 6-year period to characterize the group composition and social structure of fringe-lipped bats. Our results suggest that the majority of *T. cirrhosus* roost either in all-male groups or in single-male/multi-female groups. We found evidence for female philopatry in our long-term capture data. In addition, we found preferred co-roosting associations in both sexes, and that relatedness predicted associations among female–female dyads, but not male–male dyads.

Analyses of roost capture records over a 6-year period corroborate past evidence that fringe-lipped bats often roost in mixed-sexed groups [29–31], but we also observed single-sex groups, most often all-male groups. All-male groups always included at least one male with odorous forearm crust, they were present year-round, and the size of these groups did not appear to vary with the reproductive season. This suggests that reproductive *T. cirrhosus* males with forearm crust are not solely aggregating to perform odorous displays during the mating season, as seen for instance in the buffy flower bat, *Erophylla sezekorni* [44]. Instead, these all-male groups comprised multiple *T. cirrhosus* males with odorous forearm crust could be bachelor groups with chemical profiles that vary from males that roost with females, as seen in greater spear-nosed bats, *Phyllostomus hastatus* [45]. Most

mixed-sex groups were comprised of one reproductive male with forearm crust and several females, consistent with a single-male/multi-female mating system.

Although the spatial and temporal distribution of resources and mates are important drivers of resource or female-defense polygyny, the patterns of roost use we observed suggest that reproductive males cannot easily defend females or roosts and that roosts did not appear to be limited. Culvert roosts were situated in close proximity to one another, yet one group of bats would only occupy one culvert at a time leaving several nearby suitable roosts empty. Not only were many culverts available, but the bats often moved between tunnels even without being disturbed. For example, bats would occupy Tunnel 11 one day, then appear to move to Tunnel 12 a few days later, then move back to Tunnel 11 and then to Tunnel 9. Roost microclimate could be a factor if this varies from day to day within culverts, but a more likely explanation is that this population demonstrates fission–fusion social dynamics, like many other bats [3,4,7–9]. Furthermore, whether or not the rates of natural roost switching differ depending on the reproductive season should be explored further.

We also captured mixed-sex groups with more than two reproductive males in the group, but we did not see strong evidence that the two males were more related than expected by chance, as seen in the Jamaican fruit-eating bat, *Artibeus jamaicensis* [14,15]. In greater sac-winged bats, *Saccopteryx bilineata*, unrelated peripheral and territorial males both mate with females in the group, but territorial male *S. bilineata* are older and sire more offspring [46,47]. In these two-male/multi-female roosts, it would be interesting to investigate whether both male *T. cirrhosus* in the roost mate with females. Further studies are needed to determine the mating system in *T. cirrhosus*.

We took advantage of existing long-term capture records to infer evidence of social structure. Our sample size of repeated observations of individuals did not allow for precise descriptions or comparisons of social network structure, e.g. [4], but we detected preferred associations among both sexes and kin-biased associations among pairs of females but not males. Preferred associations occurred more than expected from a null model that accounted for year, population and shared use of roosts through the use of 'pre-network permutations' [41].

A limitation of our study was the lack of observations. Specifically, we lacked the number of repeated observations to compare social network structure with other species [4] or to test for modularity, assortativity or sex differences in centrality. Instead, we focused on what we could learn from these opportunistic long-term data. Testing for evidence of social differentiation and kin-biased association does not require highly accurate estimates of association rates between any two particular bats. One potential caveat in our analysis is missing observations of bats. Our analyses assume that missing bats were random and did not introduce biased associations between sex, kinship and association. However, even if certain types of individuals (e.g. males or females) were less likely to be captured, this bias is controlled for by our pre-network permutations, because the same biased sample exists in the permuted networks [41]. We followed the suggestion of [40] to use the simple ratio index, because more complex indices that attempt to account for missing observations are harder to interpret and require additional assumptions to reliably increase accuracy of association estimates. For example, the half-weight index does not result in better estimates of association rates unless the probability of missing an associated dyad is exactly half the probability of missing either individual in the absence of other [40]. Another caveat is that disturbances from roost captures may have led to increased movements between roosts.

In this species, recapture records were consistent with either male-biased mortality at a young age or male-biased dispersal. Even stronger evidence for male-biased dispersal comes from the finding that gene flow in fringe-lipped bats is male mediated. Male dispersal is common in mammals, including many temperate and neotropical bats [4], but females often disperse in cases where inbreeding is possible because male tenure exceeds the age of first breeding in females [5,48]. Male dispersal and female philopatry are consistent with patterns of kin-biased associations.

Kin-biased association among females can occur through kin discrimination or as a mere byproduct of females remaining in their natal group. The ability of females to recognize their adult maternal kin (e.g. mothers and daughters) is at least plausible for several reasons. In all bats studied to date, mothers appear to recognize their pups individually by scent and calls [49–54], and there is growing evidence that adult social recognition in bats is also common [54]. In *T. cirrhosus*, pups depend on maternal care for at least the first month (V.F. 2013, personal observation), which could lead to recognition of the same offspring later in life. For example, in common vampire bats *Desmodus rotundus* where females are also philopatric, mother–daughter relationships that begin as maternal care continue into adulthood as cooperative relationships that involve vocal recognition by contact call, social grooming and regurgitated food sharing [11,55–57]. Currently, there is not good evidence for recognition of unfamiliar kin in bats because studies with adequate observational or experimental power have yet to be conducted. Finally,

since females have preferred co-roosting relationships, and births are not random throughout the year [32], this could lead to further social bonding between similar age daughters (e.g. [58]).

Preferred or kin-biased associations between fringe-lipped bats could affect social learning and foraging, especially if roosts act as information centres (e.g. [59,60]). Captive *T. cirrhosus* can acquire information about novel cues via social learning [25,27], and proximity sensors show that some individuals from the same social group do encounter each other while foraging [61]. Our analysis of social networks in this species using existing records paves the way for possible field experiments that integrate social relationships and cognition, such as testing if social information transfer differs between bats that are unfamiliar versus highly associated.

Ethics. All sampling protocols followed guidelines approved by the American Society of Mammalogists for capture, handling, and care of mammals [62] and were approved by the Smithsonian Tropical Research Institute (STRI) Institutional Animal Care and Use Committee (#20100816-1012-16, #2014-0101-2017, #2017-0102-2020). All research was licensed and approved by the government of Panama (SE/A-94-11, SE/A-58-12, SE/A-19-13, SE/A-86-14, SE/A 69-15, SE/AH-2-6).

Data accessibility. Data and R code to replicate our analysis are included as electronic supplementary material.

Authors' contributions. V.F. participated in fieldwork and data collection, helped with the statistical analysis and drafted the manuscript; G.G.C. conducted the statistical analysis and helped draft the manuscript; T.K.H. provided relatedness estimates and critically revised the manuscript; G.K. organized the genetic work and critically revised the manuscript; R.A.P. organized the fieldwork and data collection and critically revised the manuscript. All authors gave final approval for publication and agree to be held accountable for the work performed therein.

Competing interests. We declare we have no competing interests.

Funding. V.F. was supported by a National Science Foundation Graduate Research Fellowship, an American Society of Mammalogists Grants-in-Aid, and a Smithsonian Tropical Research Institute Short-term Fellowship. G.G.C. was supported by a Smithsonian Postdoctoral Fellowship and a Humboldt Fellowship. Genetic analysis was funded by the Deutsche Forschungsgemeinschaft (grant no. KE 746/7-1, 7-2) within the priority program 'Ecology and species barriers in emerging viral diseases (SPP 1596)'.

Acknowledgements. We are grateful to the entire Gamboa Bat Laboratory for their long-term assistance with fieldwork. In particular, we are indebted to M. Dixon, V. Hartwell, C. Hemingway, M. Nowak, E. Ramirez and S. Vazquez for help with roost captures. The staff at the Smithsonian Tropical Research Institute provided support with logistics and permits. J. Mateo and B. Patterson provided valuable comments on this manuscript.

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
