## [Reviewer comments · Royal Society Open Science]

Review History

RSOS-192256.R0 (Original submission)

Review form: Reviewer 1

Is the manuscript scientifically sound in its present form?

Yes

Are the interpretations and conclusions justified by the results?

Yes

Is the language acceptable?

Yes

Do you have any ethical concerns with this paper?

No

Have you any concerns about statistical analyses in this paper?

No

Recommendation?

Accept with minor revision (please list in comments)

Comments to the Author(s)

This manuscripts deal with the social structure and relatedness of wild populations of *Trachops cirrhosus*. Authors performed a series of behavioral observations to assess the social structure + group composition in different roosting sites. They described in a fashion way the social arrangement using sex/age of individuals.

Authors hypothesized several predictions and used novel computational tools, such as social networks, to predict results based on their observations. They used mathematician algorithms to support their hypotheses. Results meet perfectly their expectations and they can asseverate group composition based in their predictions, also used molecular tools to assess the relatedness hypothesis in the observed structure. I have not major complaints about samples and analytical procedures, although some analyses are classical in the literature and the useful contribution can fit in other similar themes. There are too many literature based on bat social structure that is missed in the literature but no primordial for the basic content of the text. I strongly suggest to make a deeper review into bat specific literature to strengthen their introduction and discussion sections.

1.-Introduction and discussion need a better review in social structures in bat groups (wild and captive observations), because there is a lack of information regarding these papers.

2.- Figure 3 is not well represented in text, I prefer a classical graph showing mean $2) \pm SD$.

3.- Authors linked specific hypothesis with their methodology but sometimes they lack of particularities in their explanation.

Review form: Reviewer 2

Is the manuscript scientifically sound in its present form?

Yes

Are the interpretations and conclusions justified by the results?

Yes

Is the language acceptable?

Yes

Do you have any ethical concerns with this paper?

No

Have you any concerns about statistical analyses in this paper?

No

Recommendation?

Accept with minor revision (please list in comments)

Comments to the Author(s)

Reading this manuscript on the social structure of *T. cirrhosus* was a pleasure. The text is well written, and albeit data collection was opportunistic, the authors did a good job in analysing such data in providing an interesting picture on the social structure of a species poorly known in the wild.

I would only recommend the authors to discuss a bit more some aspects that may add to their interpretation of their results.

- sample size; as a bat specialist I totally understand that conducting such a study in the wild inevitably leads to small sample size, yet this point should be clearly declared, and possibly the authors may list potential biases due to this;

- roost conditions and microhabitat; authors state that roosts were not a limiting resource, yet this may be apparent only to human observers. As no microclimatic analysis was conducted on roosts, this is still a point that may have influenced the behaviour of captured bats;
 - reproductive phenology; to readers not used to bat literature (or even bat specialists not used to Latin American bat faunas) it may be useful to briefly mention the phenology of the species and its potential effects on the tests; for example: do females form nurseries where births are synchronized (if more pups are kept together within the same roost for longer periods, this may lead to the formation of social bonding too)? do reproductive females show comparable roost switching rates as non reproductive ones? Such aspects have proved important in similar studies on European bats (e.g. barbastelles: Russo et al. 2017).

Russo, D., Cistrone, L., Budinski, I., Console, G., Della Corte, M., Milighetti, C., ... & Ancillotto, L. (2017). Sociality influences thermoregulation and roost switching in a forest bat using ephemeral roosts. *Ecology and evolution*, 7(14), 5310-5321.

Decision letter (RSOS-192256.R0)

04-Mar-2020

Dear Dr Carter

On behalf of the Editors, I am pleased to inform you that your Manuscript RSOS-192256 entitled "Social structure and relatedness in the fringe-lipped bat (*Trachops cirrhosus*)" has been accepted for publication in Royal Society Open Science subject to minor revision in accordance with the referee suggestions. Please find the referees' comments at the end of this email.

The reviewers and handling editors have recommended publication, but also suggest some minor revisions to your manuscript. Therefore, I invite you to respond to the comments and revise your manuscript.

- Ethics statement

- Data accessibility

If you wish to submit your supporting data or code to Dryad (<http://datadryad.org/>), or modify your current submission to dryad, please use the following link:
<http://datadryad.org/submit?journalID=RSOS&manu=RSOS-192256>

- Competing interests

- Authors' contributions

- Acknowledgements

- Funding statement

Because the schedule for publication is very tight, it is a condition of publication that you submit the revised version of your manuscript before 13-Mar-2020. Please note that the revision deadline will expire at 00.00am on this date. If you do not think you will be able to meet this date please let me know immediately.

If your manuscript is newly submitted and subsequently accepted for publication, you will be asked to pay the article processing charge, unless you request a waiver and this is approved by Royal Society Publishing. You can find out more about the charges at <https://royalsocietypublishing.org/rsos/charges>. Should you have any queries, please contact openscience@royalsociety.org.

on behalf of Dr Alecia Carter (Associate Editor) and Kevin Padian (Subject Editor)
openscience@royalsociety.org

Associate Editor Comments to Author (Dr Alecia Carter):
Comments to the Author:

I have now received two reviews of your manuscript and read it myself. I am in agreement with the reviewers that this is a well-written and well-executed manuscript that was a pleasure to

read. In line with the reviewers' feedback that you'll find below this, I have very few comments to make:

L24: or -> and (otherwise it seems as if only one sex can do this).

L51: remove 'other'

Results section (and elsewhere): the mean is not an appropriate average to report for count data, which are not normally distributed. Please report the median (not the mean) and IQR (not the SD) throughout.

L240: harder -> more difficult (but this is just my preference)

LL241-242: Please report the analyses that support this assertion (perhaps in supplementary material)

L271: are odorous displays common? This is a bit out-of-the-blue, and could do with some contextualisation.

Discussion: Although the entire manuscript is well written, generally I prefer more flagging in the discussion, and a beginning paragraph that re-introduces the main questions and findings before discussing specifics. But I am aware that this is a personal preference. In this case, though, I found that the discussion jumps right in and it took me some time to catch up with how what was being reported was related to the study's aims, and I would encourage the authors to flag the questions / re-state their predictions to make it easier for the reader to follow throughout.

Reviewer comments to Author:

Reviewer: 1

Comments to the Author(s)

This manuscripts deal with the social structure and relatedness of wild populations of *Trachops cirrhosus*. Authors performed a series of behavioral observations to assess the social structure + group composition in different roosting sites. They described in a fashion way the social arrangement using sex/age of individuals.

Authors hypothesized several predictions and used novel computational tools, such as social networks, to predict results based on their observations. They used mathematician algorithms to support their hypotheses. Results meet perfectly their expectations and they can asseverate group composition based in their predictions, also used molecular tools to assess the relatedness hypothesis in the observed structure. I have not major complaints about samples and analytical procedures, although some analyses are classical in the literature and the useful contribution can fit in other similar themes. There are too many literature based on bat social structure that is missed in the literature but no primordial for the basic content of the text. I strongly suggest to make a deeper review into bat specific literature to strengthen their introduction and discussion sections.

- 1.-Introduction and discussion need a better review in social structures in bat groups (wild and captive observations), because there is a lack of information regarding these papers.
- 2.- Figure 3 is not well represented in text, I prefer a classical graph showing mean \pm SD.
- 3.- Authors linked specific hypothesis with their methodology but sometimes they lack of particularities in their explanation.

Reviewer: 2

Comments to the Author(s)

Reading this manuscript on the social structure of *T. cirrhosus* was a pleasure. The text is well written, and albeit data collection was opportunistic, the authors did a good job in analysing such data in providing an interesting picture on the social structure of a species poorly known in the wild.

I would only recommend the authors to discuss a bit more some aspects that may add to their interpretation of their results.

- sample size; as a bat specialist I totally understand that conducting such a study in the wild inevitably leads to small sample size, yet this point should be clearly declared, and possibly the authors may list potential biases due to this;
- roost conditions and microhabitat; authors state that roosts were not a limiting resource, yet this may be apparent only to human observers. As no microclimatic analysis was conducted on roosts, this is still a point that may have influenced the behaviour of captured bats;
- reproductive phenology; to readers not used to bat literature (or even bat specialists not used to Latin American bat faunas) it may be useful to briefly mention the phenology of the species and its potential effects on the tests; for example: do females form nurseries where births are synchronized (if more pups are kept together within the same roost for longer periods, this may lead to the formation of social bonding too)? do reproductive females show comparable roost switching rates as non reproductive ones? Such aspects have proved important in similar studies on European bats (e.g. *barbastelles*: Russo et al. 2017).

Russo, D., Cistrone, L., Budinski, I., Console, G., Della Corte, M., Milighetti, C., ... & Ancillotto, L. (2017). Sociality influences thermoregulation and roost switching in a forest bat using ephemeral roosts. *Ecology and evolution*, 7(14), 5310-5321.

Author's Response to Decision Letter for (RSOS-192256.R0)

See Appendix A.

Decision letter (RSOS-192256.R1)

19-Mar-2020

Dear Dr Carter,

It is a pleasure to accept your manuscript entitled "Social structure and relatedness in the fringe-lipped bat (*Trachops cirrhosus*)" in its current form for publication in Royal Society Open Science. The comments of the reviewer(s) who reviewed your manuscript are included at the foot of this letter.

Due to rapid publication and an extremely tight schedule, if comments are not received, your paper may experience a delay in publication. Royal Society Open Science operates under a continuous publication model. Your article will be published straight into the next open issue and this will be the final version of the paper. As such, it can be cited immediately by other

researchers. As the issue version of your paper will be the only version to be published I would advise you to check your proofs thoroughly as changes cannot be made once the paper is published.

on behalf of Dr Alecia Carter (Associate Editor) and Kevin Padian (Subject Editor)
openscience@royalsociety.org

Appendix A

Ref.: RSOS-192256

Title: Social structure and relatedness in the fringe-lipped bat (*Trachops cirrhosus*)

March 12, 2020

Dear Dr. Carter,

Thank you for considering our manuscript for publication in *Royal Society Open Science*. We are grateful for the helpful comments and critiques provided by both the reviewers. We have incorporated all the reviewer recommendations, and think that the manuscript is stronger and clearer as a result. We respond to each individual point in **bold** below.

We thank you very much and look forward to hearing from you soon.

With best wishes,

Gerald Carter
On behalf of all authors

Associate Editor Comments to Author (Dr Alecia Carter):

Comments to the Author:

I have now received two reviews of your manuscript and read it myself. I am in agreement with the reviewers that this is a well-written and well-executed manuscript that was a pleasure to read. In line with the reviewers' feedback that you'll find below this, I have very few comments to make:

L24: or -> and (otherwise it seems as if only one sex can do this).

Thank you pointing this out. The text now reads:

to infer whether male and female *T. cirrhosus* have preferred co-roosting associations and whether such associations were influenced by relatedness

L51: remove 'other'

Sentence now reads:

In addition, dispersal by one sex can lead to kin-biased associations in the philopatric sex

Results section (and elsewhere): the mean is not an appropriate average to report for count data, which are not normally distributed. Please report the median (not the mean) and IQR (not the SD) throughout.

We appreciate you pointing this out. We report the median and IQR. In addition, we also report the mean difference when we used it in our permutation tests (which do not assume a normal distribution).

L240: harder -> more difficult (but this is just my preference)

Sentence now reads:

We confirmed that social differentiation was more difficult to detect when including more bats that were only observed a few times.

L241-242: Please report the analyses that support this assertion (perhaps in supplementary material)

This line now reads:

When not excluding bats seen only a few times, we could only clearly detect social differentiation in females (137 adults: $CV = 5.45$, $p = 0.2$; 61 adult females: $CV = 4.24$, $p = 0.004$, 76 adult males: 6.13 , $p = 0.6$).

We also added this text to the methods

We only include bats observed at least four times in our final analyses ($n = 34$), but we also report the results when including all bats.

L271: are odorous displays common? This is a bit out-of-the-blue, and could do with some contextualisation.

Males in several bat species produce odorous displays. However, what is not common is a group of males aggregating to court females. We have re-written this section to provide some clarification:

All-male groups always included at least one male with odorous forearm crust, they were present year-round, and the size of these groups did not appear to vary with the reproductive season. This suggests that reproductive *T. cirrhosus* males with forearm crust are not solely aggregating to perform odorous displays during the mating season, as seen for instance in the buffy flower bat, *Erophylla sezekorni* (45). Instead, these all-male groups comprised of multiple *T. cirrhosus* males with odorous forearm crust could be bachelor groups with chemical profiles that vary from males that roost with females, as seen in greater spear-nosed bats *Phyllostomus hastatus* (46). Most mixed-sex groups were comprised of one reproductive male with forearm crust and several females, consistent with a single-male/multi-female mating system.

Discussion: Although the entire manuscript is well written, generally I prefer more flagging in the discussion, and a beginning paragraph that re-introduces the main questions and findings before discussing specifics. But I am aware that this is a personal preference. In this case, though, I found that the discussion jumps right in and it took me some time to catch up with how what was being reported was related to the study's aims, and I would encourage the authors to flag the questions / re-state their predictions to make it easier for the reader to follow throughout.

Thank you for this comment. We have added a paragraph summarizing our study at the beginning of the discussion:

In this study, we used data from bats captured in their roosts over a 6-year period to characterize the group composition and social structure of fringe-lipped bats. Our results suggest that the majority of *T. cirrhosus* roost either in all-male groups or in single-male/multi-female groups. We found evidence for female philopatry in our long-term capture data. In addition, we found preferred co-roosting associations in both sexes, and that relatedness predicted associations among female-female dyads, but not male-male dyads.

Reviewer comments to Author:

Reviewer: 1

Comments to the Author(s)

This manuscripts deal with the social structure and relatedness of wild populations of *Trachops cirrhosus*. Authors performed a series of behavioral

observations to assess the social structure + group composition in different roosting sites. They described in a fashion way the social arrangement using sex/age of individuals.

Authors hypothesized several predictions and used novel computational tools, such as social networks, to predict results based on their observations. They used mathematician algorithms to support their hypotheses. Results meet perfectly their expectations and they can asseverate group composition based in their predictions, also used molecular tools to assess the relatedness hypothesis in the observed structure. I have not major complaints about samples and analytical procedures, although some analyses are classical in the literature and the useful contribution can fit in other similar themes. There are too many literature based on bat social structure that is missed in the literature but no primordial for the basic content of the text. I strongly suggest to make a deeper review into bat specific literature to strengthen their introduction and discussion sections.

1.-Introduction and discussion need a better review in social structures in bat groups (wild and captive observations), because there is a lack of information regarding these papers.

We appreciate this comment, and we have expanded the section on bat social structure in the introduction which now reads:

Many bat species demonstrate nonrandom social structure including stable social relationships persisting for years, despite frequent roost switching and fission-fusion dynamics [reviewed by (4,7,8)]. For example, preferred associations are evident in philopatric female vampire bats, *Desmodus rotundus* (9–11), female *Bechstein's* bats, *Myotis bechsteinii* (3,12,13), male *Jamaican fruit-eating bats*, *Artibeus jamaicensis* (14,15), and both male and female *Spix's disk-winged bats*, *Thyroptera tricolor* (16,17). In all these species, individuals switch between a number of roosts, but emergent social structures can be detected by observing marked individuals over time (4). In bats, reproduction is slow (typically one or two pups a year) but lifespans can be quite long (up to 40 years), one or both of the sexes typically disperse at sexual maturity, and kinship within groups typically has a low mean and high variance (4,8,12,18–23). Taken together, these factors create opportunities for both long-term and kin-biased social relationships, but the degree of kinship bias in association rates varies greatly between species (4). In many bats, kinship does not seem to be a key driver of co-roosting associations or even cooperative interactions (4,8,12).

2.- Figure 3 is not well represented in text, I prefer a classical graph showing mean \pm SD.

We now refer to Figure 3 when reporting the juvenile recapture times (line 245). We added a violin plot around the raw data to highlight the distribution. The point of this plot is that there is greater variation in

recapture periods for the females and that males are never re-captured after around 700 days. A graph of the means and standard deviations would obscure these points, which is why we used a violin plot along with the raw data.

Figure 3. Juvenile males were never recaptured as older adults. Females show greater variation in time between the first capture as a juvenile and last capture as an adult ($F = 0.15$, $n = 25$, $p < .0001$).

3.- Authors linked specific hypothesis with their methodology but sometimes they lack of particularities in their explanation.

For more clarity, we now summarized our results in the discussion. See response to editor comments above. We also made some minor grammatical changes for clarity, such as moving text within a sentence.

Reviewer: 2

Comments to the Author(s)

Reading this manuscript on the social structure of *T. cirrhosus* was a pleasure. The text is well written, and albeit data collection was opportunistic, the authors did a good job in analysing such data in providing an interesting picture on the social structure of a species poorly known in the wild.

I would only recommend the authors to discuss a bit more some aspects that may add to their interpretation of their results.

- sample size; as a bat specialist I totally understand that conducting such a study in the wild inevitably leads to small sample size, yet this point should be clearly declared, and possibly the authors may list potential biases due to this;

We added the following text to the discussion:

A limitation of our study was the lack of observations. Specifically, we lacked the number of repeated observations to compare social network

structure with other species (4) or to test for modularity, assortativity, or sex differences in centrality. Instead, we focused on what we could learn from these opportunistic long-term data. Testing for evidence of social differentiation and kin-biased association does not require highly accurate estimates of association rates between any two particular bats.

- roost conditions and microhabitat; authors state that roosts were not a limiting resource, yet this may be apparent only to human observers. As no microclimatic analysis was conducted on roosts, this is still a point that may have influenced the behaviour of captured bats;

We added the following text to the discussion:

Not only were many culverts available, but the bats often moved between tunnels even without being disturbed. For example, bats would occupy Tunnel 11 one day, then appear to move to Tunnel 12 a few days later, then move back to Tunnel 11 and then to Tunnel 9. Roost microclimate could be factor if this varies from day to day within culverts, but a more likely explanation is that this population demonstrates fission-fusion social dynamics, like many other bats (3,4, 7-9).

- reproductive phenology; to readers not used to bat literature (or even bat specialists not used to Latin American bat faunas) it may be useful to briefly mention the phenology of the species and its potential effects on the tests; for example: do females form nurseries where births are synchronized (if more pups are kept together within the same roost for longer periods, this may lead to the formation of social bonding too)? do reproductive females show comparable roost switching rates as non reproductive ones? Such aspects have proved important in similar studies on European bats (e.g. barbastelles: Russo et al. 2017).

Russo, D., Cistrone, L., Budinski, I., Console, G., Della Corte, M., Milighetti, C., ... & Ancillotto, L. (2017). Sociality influences thermoregulation and roost switching in a forest bat using ephemeral roosts. *Ecology and evolution*, 7(14), 5310-5321.

We don't yet know much about reproduction in this species, and this is essentially the first study on its social structure. We have added the following sentences.

Whether or not the rates of natural roost switching differ depending on the reproductive season should be explored further.

Finally, since females have preferred co-roosting relationships, and births are not random throughout the year (32), this could lead to further social bonding between similar age daughters (e.g. 59).

We also added this citation:

Ancillotto L, Serangeli MT, Russo D. Spatial proximity between newborns influences the development of social relationships in bats. *Ethology*. 2012;118(4):331-40.